# Remote Sensing Image Dataset Expansion Based on Generative Adversarial Networks with Modified Shuffle Attention

**DOI:** 10.3390/s21144867

**Published:** 2021-07-16

**Authors:** Lu Chen, Hongjun Wang, Xianghao Meng

**Affiliations:** School of Electronic Countermeasures, National University of Defense Technology, Hefei 230000, China; chenlu20@nudt.edu.cn (L.C.); mengxianghao17@nudt.edu.cn (X.M.)

**Keywords:** modified shuffle attention GAN, limited input dataset, picture generation, resize method

## Abstract

With the development of science and technology, neural networks, as an effective tool in image processing, play an important role in gradual remote-sensing image-processing. However, the training of neural networks requires a large sample database. Therefore, expanding datasets with limited samples has gradually become a research hotspot. The emergence of the generative adversarial network (GAN) provides new ideas for data expansion. Traditional GANs either require a large number of input data, or lack detail in the pictures generated. In this paper, we modify a shuffle attention network and introduce it into GAN to generate higher quality pictures with limited inputs. In addition, we improved the existing resize method and proposed an equal stretch resize method to solve the problem of image distortion caused by different input sizes. In the experiment, we also embed the newly proposed coordinate attention (CA) module into the backbone network as a control test. Qualitative indexes and six quantitative evaluation indexes were used to evaluate the experimental results, which show that, compared with other GANs used for picture generation, the modified Shuffle Attention GAN proposed in this paper can generate more refined and high-quality diversified aircraft pictures with more detailed features of the object under limited datasets.

## 1. Introduction

Remote Sensing (RS) refers to the non-contact remote detection technology. Images obtained by remote sensing technology cover a large range of landforms and features, which contain a lot of information. Remote-sensing image interpretation is used to judge the natural landform, artificial terrain, and target information of the RS images, which is widely used in civil fields such as ground feature coverage [1,2,3] and forest detection [4].

RS images often have a huge amount of information while the targets are difficult to observe, which brings a great challenge and heavy burden to traditional manual interpretation. Therefore, the selection and processing of RS images by computer have become a research hotspot of RS image interpretation [5,6,7]. The machine interpretation of RS images mainly consists of the detection and classification of terrain, landform, and target [8,9]. With the rapid development of neural networks in the field of natural image processing, the method of extracting the depth characteristics has also been applied in the processing of RS images to maintain better performance.

Most target detection methods of neural networks are aimed at natural image datasets. At present, compared with the natural image dataset, the RS image dataset still covers fewer target types and amounts. Consequently, it is necessary to expand the dataset of a remote-sensing image. However, due to the high labor cost of labeling sample sets, it is more difficult to find targets in a large number of remote-sensing images and carry out manual labeling. Sample expansion on the existing limited datasets has become an important method to solve the problem of insufficient samples. The traditional expansion methods include clipping, mirroring, rotation, and other highly coupled expansion methods. Nowadays, we are faced with the problem: “How can one generate abundant and multiple images with a limited and unlabeled database?”.

In 2014, Goodfellow et al. proposed generative adversarial networks (GANs) [10], which provide a new method of image generation [11,12]. A conditional generation network (CGAN) [13] adds conditional constraints to the GAN of unsupervised learning and guides the data generation process by adapting the model to additional information, which is also applied for better recognition accuracy [14]. However, it requires a labeled-images database. LAPGAN [15] uses the cascading of convolutional networks within the framework of the Laplacian pyramid to generate coarse to fine images, which is usually used in target detection rather than sample generation.

The local convolution in the GAN network usually needs to go through many convolution layers to connect the distant features that belong to different regions, and the increase of the convolution kernel will significantly reduce the computational and statistical efficiency of network training. Considering the condition above, SAGAN [16] introduces the self-attention mechanism into the GAN network. To balance the relationship between long-distance dependence and computational efficiency, the self-attention module calculates the response at a certain location as a weighted sum of all location features, where the weights can be calculated with only a small computational cost. SaGAAN [17] introduces the self-attention block into the generative adversarial adaptation network (GAAN) to generate the labeled samples for image classification, which can greatly improve the classification accuracy. VSA-CGAN [18] integrates the SAGAN network and the CGAN network and solves the problem of unsupervised learning in the SAGAN network. However, the VSA-CGAN network still requires the labeled image database. CBAM-GAN [19] introduces the convolutional block attention Module (CBAM) [20] into GAN in order to improve the generated pictures.

In these GANs mentioned above, SAGAN and CBAM-GAN proposed an appropriate method in picture generation based on a limited unlabeled image database. However, the self-attention module applied in SAGAN belongs to the spatial attention module. It neglects the influence of channel attention. CBAM-GAN connects the two modules in series, but its pooling and multilayer perceptions need to be hand-designed. Therefore, we aim to introduce a more efficient and effective attention module into GAN to solve the problem of image expansion.

Based on the problem mentioned above, this paper carried out research, and the contributions of this paper are as follows:We adjusted the structure of the shuffle attention network by replacing the original spatial attention module with a modified self-attention network to obtain better spatial attention. Then we proposed a modified shuffle attention GAN by introducing the modified shuffle attention net into GAN.We introduced the mini-batch into the backbone network to avoid mode collapse in GAN under a small dataset. In the later experiment, the 1-NN index verified the effect brought about by this introduction.We introduced a coordinate attention network into GAN to form coordinate attention GAN in contrast with the shuffle attention GAN. We integrate the civil aircraft in the dataset NWPUVHR-10 and UC Merced Land use and select the input pictures to ensure the purity of the input type of aircraft.We proposed an equal stretch resize method to avoid the distortion of the image. We also conducted contrast experiments on the modified shuffle attention GAN with and without the equal stretch resize method. The results shows that the equal stretch resize method can apparently avoid the distortion.We applied a qualitative and quantitative evaluation index to judge the quality of the generated pictures. The qualitative evaluation consists of the appearance of the fuselage, the amount and location of the engine, and the symmetry of the aircraft. The representative quantitative evaluation index consists of an inception score (IS), The Fréchet inception distance (FID) [21], the mode score, the Kernel Maximum Mean Discrepancy (MMD), the Wasserstein distance (WD), and 1-Nearest Neighbor Classifier (1-NN). We make comparisons and conduct an evaluation on the results of the modified shuffle attention GAN and the existing image generation GAN. The experiments show that the modified shuffle attention GAN performed better than GAN, SAGAN, CBAM-GAN, and coordinate attention GAN.

The rest of this paper is organized as follows. We introduce the basic structure of GAN in Section 2. In Section 3, we illustrate the structure of the modified shuffle attention GAN and the equal stretch resizing method. In Section 4, we illustrate the pretreatment of the database and the evaluation index to judge the quality of the generated picture. In Section 5, we display, evaluate, and analyze the results of the modified shuffle attention GAN and other GAN modules with and without the equal-stretch resizing method. Finally, Section 6 summarizes the whole paper and gives a conclusion. 

## 2. Related Works

### 2.1. Generative Adversarial Network

Generative adversarial networks (GANs) are a generative unsupervised learning model, which is mainly inspired by the idea of a zero-sum game. Figure 1 is the basic structure diagram of GANs. 

GAN consists of two modules: A generator and a discriminator. The generator changes the input random noise vector z into a picture of the same size as the sample, and the input of the discriminator is the alternate sample and the fake sample generated by the generator, which is classified through the network. If the discriminator determines that the false sample is false, the error is transmitted back to the generator to update the generator network. At the same time, the classification error obtained from each training will be transmitted back to the discriminator to constantly update the discriminator network. The generator and discriminator are constantly updated in the training until the “Nash Equilibrium” is reached. In article [22], the criterion for reaching the Nash Equilibrium is that the loss of the discriminator fluctuates around 0.5.

D represents discriminator while G is the generator, and V is the value function of GAN. z represents the random noise input into the generator, while x represents the image from the dataset. The value function of GAN is expressed as:(1)$$\underset{G}{\min}\;\underset{D}{\max}\;V(D,G)=E_{x\sim p_{data}(x)}\left[\log D(x)\right]+E_{z\sim p_{z}(z)}\left[\log(1-D(G(z)))\right]$$

The value function can be split into two parts for the discriminator and the generator, representatively. For one side, we expect the D (discriminator) to identify fake and true, and for the other side, we expect to generate more similar fake pictures to confuse the discriminator.

Through a continuous game between the generation network and the discriminant network, fake samples can be generated similarly to the real sample. Generative adversarial networks have recently been introduced as an alternative framework for training generation models to sidestep many of the intractable difficulties of probabilistic computation approximation. The advantage of an adversarial net is that it does not need a Markov chain and only needs backpropagation to obtain the gradient. It does not need reasoning in the learning process and can easily incorporate various factors and interactions into the model.

### 2.2. Attention Module

The attention module aims to focus attention on the region of interest (ROI) in the network, which has been applied in various computer vision tasks, such as image classification [23,24,25] and image segmentation [26,27,28,29,30,31]. In the last few years, the majority of the research on the combination of deep learning and the visual attention module has focused on using masks to form the attention module. The principle of the mask is to identify the key features of input pictures through another layer of new weight. Through learning and training, the weights given keep updating and gradually differ from each other. The larger the trained weight is, the more important the corresponding feature is. Therefore, the neural network learns to focus on the region that needs to be paid attention to in every input picture. After training, the weight will be combined with inputs. There are different ways of combination, which decides the type of attention module. The channel attention module and spatial attention module are two widely used attention Module.

In a convolutional neural network, pictures consist of an initial three channels—R, G, and B. After different convolutional kernels, every original channel will generate a new signal. A convolutional kernel is used as a kind of transformation to make different processing changes to the original signal. Therefore, each channel represents a new meaning compared with the three channels after convolution. The receptive field of the convolution kernel is local, which must be accumulated through many layers before the regions of different parts of the whole image can be associated. Therefore, SE [32] appears in CVPR 2018, which measures the global information of images from the level of feature channels. SE Net is the first to come up with the Squeeze-and-Excitation model from the Chanel-wise level. ECA-Net [33] is an improvement of SE, which adopted a 1-D convolution filter to generate channel weights and significantly reduced the model complexity of SE.

Channel attention gives weight to the signals on each channel, representing how relevant that channel is to the key information. The higher the weight is, the higher the correlation is and the more channel attention should be paid. Channel attention uses the attention module to learn the weight of each channel in the process of network training, so as to highlight the contribution of signals with a large amount of information in the whole feature map.

Different from channel attention, spatial attention focuses on the region that contributes more to the key information. The mask is combined with spatial attention rather than channel attention. In the processing, the same position of all channels needs to be averaged or convoluted in order to ensure the feature map of post-processing synthesizes the information of all channels. Wang et al. [34] proposed the non-local (NL) module to capture long-range dependencies with the response at a position of the weighted sum of the features in all positions.

In order to realize channel and spatial attention at the same time, some networks combined the two attention modules and achieved significant improvement. GC-Net [35] integrated two attention mechanisms into one module but faced converging difficulty. SGE [36] divided the dimensions of the channel into multiple sub-features to learn different semantics but failed to take full advantage of the correlation between spatial and channel attention. DA-Net [26] built a parallel attention mechanism of PAM and CAM on the traditional expanded FCN. The structure of PAM is similar to that of NL in that the dependency relationship between features is obtained by calculating the correlation graph of the feature graph. CAM also adopts this idea and constructs a correlation graph on the channel to represent the dependency relationship between any two channels. This avoids the complex operation of manual pooling and multi-layer perceptron design in CBAM, but it will require a large amount of computation to determine the weight of the channel or the feature by building the dependency relationship. Recently, the coordinate attention (CA) [37] was proposed to embed the location information into the channel attention system, which can capture the long-range correlation along one spatial direction, while preserving the exact location information along the other spatial direction. CA-Net performed better than CBAM and SE-Net in the experiment. However, CA-Net has an unstable performance when being integrated into GAN in our experiment.

Although the immigrations of both spatial attention and channel attention perform better than the single attention module, the algorithm complexity is inevitably increased. In order to balance the computation efficiency and resulting effectiveness, a shuffle attention net (SA-Net) was proposed [38] to combine the two attention modules efficiently. 

## 3. Modified Shuffle Attention GAN

### 3.1. Backbone of Shuffle Attention GAN

Like the basic structure of GAN, the shuffle attention GAN is divided into a discriminator and a generator.

The specific network structure of the discriminator and the generator is shown in Figure 2a. Firstly, the random noise Z vector is input to the full connection layer through the full connection layer, and the length of the noise Z is initially set as an exponential time of 2 to facilitate subsequent upsampling-ResNet and other subsequent operations. Among them, upsampling-ResNet adds upsampling on the basis of the structure of the residual network. On the basis of retaining the original features, it can also refine the details generated by the generator feature map through the interpolation method. Paper [39] illustrates that the order of BN, ReLU, and weight achieved the fastest error reduction and lowest training loss. Therefore, here in the upsampling-ResNet, we designed in the order of BN, ReLU, and up-sampling. Moreover, it has also been proved that the structure of the original shortcut of ResNet performed better than the shortcut impeded by different components. Here we only put up-sampling in the shortcut.

The number of layers in the upsampling-ResNet is set as $\left|L/2\right|+1$, where $L=\left(log2\left(\max(H,W)\right)\right)-3$. Before each ReLU activation function is processed, the Batch Normal module is added for normalization, that is, the mean value is subtracted, and the variance is unitized. Batch Normal can reduce the absolute difference between weights, highlight the relative difference of weights, and accelerate the learning rate of the network. 

We added the attention module between two groups of upsampling-ResNet. The output of the first group of upsampling-ResNet was used as the feature maps of the attention module. The feature map processed by the attention module pays more attention to the feature regions and channels related to the task objectives. After that, the feature map was further refined by upsampling-ResNet processing.

Finally, the size of the generated graph was adjusted through the 3×3 convolutional network and the final pseudo sample was output through the Tanh activation function.

Similarly, the discriminator alternately takes false and real samples as inputs. The input sample first obtains feature maps through a downsampling-ResNet. Similar to a generator’s upsampling-ResNet, a down-residual network is added to a residual network block by subsampling. The feature map obtained after the first group of the down-residual network is entered by the sample will also serve as the input of the attention module. The attention module of the discriminator is the same as the generator, so that the discriminator and generator can pay the same attention to the feature graph.

In the discriminator, all activation functions are Leaky ReLU. In contrast to the ReLU function, Leaky ReLU does not always have a zero output on the negative side of the axis, which has a very small slope and thus allows neurons to update despite a negative input. The second downsampling-ResNet of the discriminator has a number of levels L. 

Mode collapse is a common problem for GAN models. Mode collapse happens while the generator learns how to cheat the discriminator with a small group of generated images. Generally, mode collapse is closely related to the discriminator. The discriminator can only process one sample independently at a time, which contributes to the lack of information coordination between samples. The mini-batch discriminator is an effective solution to avoid mode collapse [22]. The simplified version of a mini-batch [40] was added after the last down-ResNet layer to avoid mode collapse. The details of the simplified mini-batch are as follows:(2)$$o=\frac{1}{n}\sum_{i=1}^{n}\left(\sigma_{i}\right)$$
(3)$$\sigma_{i}=\sqrt{\frac{1}{m-1}\sum_{j=1}^{m}(f{\left(x_{j}\right)}_{i}-{\hat{f}}_{i}{)}^{2}}$$

$x_{j}$ represents the input samples of the discriminator and $f{\left(x_{j}\right)}_{i}$ represents the *i*-dimension feature of the sample $x_{j}$. $\sigma_{j}$ is the standard deviation of *i*-dimension. The result of o will be combined with the output of the last downsampling-ResNet layer. In this way, the discriminator can learn the features of a group of samples rather than a single sample, which can effectively avoid mode collapse.

After the mini-batch, the input samples were identified by global summation pooling and full connection layer, and the prediction category $\bar{D}$ of the discriminant model was obtained.

The loss function is shown above. y represent the attention mask of a data feature map, while z represents the random noise.
(4)$$L_{G}=-E_{z\sim P_{z},y\sim P_{data}}D(G(z),y))$$
(5)$$L_{D}=-E_{(x,y)\sim P_{data}}[\min(0,-1+D(x,y))]-E_{z\sim P_{z},y\sim P_{data}}[\min(0,-1-D(G(z),y))]$$

### 3.2. Structure of Attention Module

The discriminator and generator have the same “shuffle attention” module. The shuffle attention block combines channel attention and spatial attention. In this paper, the spatial attention module in shuffle attention is replaced by the self-attention module, and channel attention remains the structure in the shuffle attention network.

The structure of shuffle attention is shown in Figure 3. Suppose the shape of the feature map is [N,C,H,W], with N for the number of the batch size, C for the channel, H for height, and W for width. The feature map is divided into several groups in the dimension of the channel, and the shape of each group is [N,C/G,H,W].

For each group ($g_{i}$), $g_{i}$ is split into $x_{k1}$ and $x_{k2}$ where one branch produces a channel attention map and the other produces a spatial attention map. In order to combine the two feature maps, masks should keep the same shape. Then the combination of the two spatial attention maps will go through the activator to form the weight of each group. After the aggregation, feature maps will be shuffled in the dimension of the channel to realize information communication between different sub-features.

#### 3.2.1. Channel Attention

Figure 4 displayed the structure of SE-Net, ECA-Net, and original channel attention applied in the shuffle attention network.

SE-Net first transforms each two-dimensional feature channel into a real number through global average pooling ($F_{gp}(•)$). By using this method, the problem of a lack of a global receptive field in the convolutional layer can be solved, and the global distribution on the characteristic channel can be characterized. To relieve the computational burden, SE then compresses the number of channels through a full connection layer. The function is activated by ReLU, and then connected to the full connection layer again to restore the original channel number. By scaling, the mask is overlayed on the initial feature map to obtain channel attention.

In the first step of ECA, the global average pooling method is also applied to characterize the global distribution of channels. Unlike SE, ECA does not reduce the dimension in the post-processing process, but outputs the mask directly through the full connection layer and a convolution layer. The results of the ECA experiment show that avoiding dimensionality reduction helps to learn effective channel attention. In addition to the local cross-information interaction strategy with no dimensionality reduction, ECA also adds a convolution that adaptively selects the size of the convolution kernel to ensure that the mask dimension of the output in the later period can match the channel number.

Here, Figure 4c displays channel attention of shuffle attention. The channel attention module in shuffle attention uses global pooling to embed global information in the first step. $x_{k1}$ represents the branch of $g_{i}$.
(6)$$z=F_{gp}(x_{k1})=\frac{1}{H\times W}\sum_{i=1}^{H}\sum_{j=1}^{W}x_{k1}(i,j)$$

By means of Global Pool, features in the same channel are added and averaged. z is a representative of information of $x_{k1}$.
(7)$$x_{{}_{k1}}^{\prime }=\sigma (W_{1}z+b_{1})•x_{k1}$$

By learning these two weights through training, a one-dimensional incentive weight is obtained to activate each layer channel. $\sigma (W_{1}z+b_{1})$ acts as the weight of channels, the core of the channel attention module. $W_{1}$ is initialized as zero, while $b_{1}$ is initialized as one. The value of $W_{1}$ and $b_{1}$ is constantly updated with the backpropagation error.

Without dimensionality reduction, channel attention can be fully learned. Besides, it gets rid of the adaptive convolution of ECA in the way of adding bias, which further reduce the parameters and relieves the computation burden.

#### 3.2.2. Spatial Attention

In the experiment we found that, with the same training iteration and same training database, it is difficult to identify the fake pictures GAN generated, while the fake pictures SAGAN generated already have highly recognizable aircraft features. Relevant experimental results are presented in Section 5. Since the addition of the SA module in SAGAN can significantly improve the image generation of the GAN network. In order to continue the good performance of the self-attention block in spatial attention, we replace the original attention block in the shuffle attention network with a modified self-attention block.

Furthermore, since the product of g(x) and f(x) needs to be calculated in self attention, when the input feature map is large, the amount of calculation will increase. In order to reduce the amount of calculation and ensure the accuracy of the attention region as much as possible, we only add a maximum pooling to the branch of convolutional results of x, as it is shown in Figure 3.

The feature map x∈ℝC×N is the result of the previous convolution. Firstly, self-attention uses 1×1 convolution to compress the number of channels in order to integrate channel information. With different 1×1 convolution, we get different reflections of x: f(x)=Wfx, g(x)=Wgx, h(x)=Whx. Moreover, the channel number of f(x) and g(x) are squeezed into ch/8 in order to mix cross-channel information and reduce the amount of late computation. Fmp(•) represents the max pooling.
(8)sij=Fmp[f(xi)]Tg(xj)

$s_{ij}$ is the product of the feature map f(x) and g(x). In the initial 1×1 convolution, f(x) and g(x) use the same convolution kernel and get the same feature map. Therefore, $s_{ij}$ can be regarded as the autocorrelation matrix of the feature map, which represents the correlation of each pixel to the pixel of the entire feature map. $s_{ij}$ is the core to build spatial attention.
(9)$$\beta_{j,i}=\frac{\exp(s_{ij})}{\sum_{i=1}^{N}\exp(s_{ij})}$$

$\beta_{j,i}$ indicates the model attention i location while synthesizing region j.
(10)$$o_{j}=v(\sum_{i=1}^{N}\beta_{j,i}h(x_{i})),\;v(x_{i})=W_{v}x_{i}$$

v(x) is the mask of the feature map. It should be added into the input to change the weights of each feature. SAGAN also multiplies a scale parameter γ to control the influence of the mask.
(11)$$y_{i}=\gamma o_{i}+x_{i}$$

### 3.3. Promoting Method of Picture Resizing

In the process of network training, there is a unified size processing for samples in the initial processing, among which the resize function in Python directly stretches the original image to a certain proportion. This will distort the original sample aircraft image due to the different sizes of the sample aircraft image input. The proportions of the aircraft are destroyed, the generator cannot learn the features of the aircraft in normal proportions, and the resulting fake sample will become distorted with the distortion of the sample.

Therefore, when preprocessing the sample dataset, we replaced the original resize function and proposed a new method to resize the pictures, that is, the equal stretch resize. The specific operation is shown in Figure 5.

The input image has to be square, and the length and width has to be the power of 2. Firstly, we fill along the short side based on the long side, setting the values to 255 for all areas that need to be filled. Then the completed square is constructed by stretching in equal proportion to get the final sample of the target length.

In this way, the distortion caused by direct stretching can be reduced, and because the information contained by the white edge is small, similarly to the information brought by the background, the existence of the white edge will not have an impact on the generation of aircraft fake samples.

## 4. Experiments

This section will illustrate the pretreatment of the database and illustrate the necessity of the pretreatment. Training details consist of the training environment, parameters, and control group settings. A qualitive and quantitative evaluation index will also be constructed and explained in Section 4.

### 4.1. Database and Pretreatment 

The dataset NWPUVHR-10 and UC Merced Land-use were used to construct the aircraft dataset, and most of the aircraft in the dataset were civil aircraft. Civil aircraft have single swept-back wings and typically have two or four turbojet or turbofan engines located below the wings. Although civil aircraft are also divided into large passenger aircraft and small passenger aircraft, the size difference is not very obvious, and the overall shape and proportion of the aircraft are similar. In addition to commercial aircraft, there are other aircraft in the aircraft database as shown in the figure.

The input sample image determines the fake sample generated by the generated network. If the input sample image contains various types of aircraft or there is a large difference between the shapes of aircraft, it will lead to the generation of network imitation of the fake sample image combining various aircraft features. What can be seen here is a fake sample generated by the input of different types of aircraft.

Such samples of the aircraft displayed in Figure 6c do not exist, and the resulting fake sample images are not useful. If the sample database is used for other network training, such as aircraft classification and target detection, errors will occur at the source. Therefore, it is necessary to screen the input sample data. The databases used in this paper are all civil aircraft with obvious characteristics.

The original image is an RGB three-channel color image, but most of the body of the actual aircraft is white. The RGB color sample image does not contribute much to the identification of aircraft features. In order to reduce the amount of computation, the color image is transformed by HSV, and the sample image of the V channel is taken as the input sample, that is, the original image is grayscale processed.

### 4.2. Training Details

We built the network with Python version 3.7.3 and TensorFlow version 1.13.1 in GPU. We trained the network on a desktop computer whose host video card is RTX2070 SUPER.

In order to further illustrate the improvement of the GAN network brought by the insertion of shuffle attention, we set up several sets of comparative experiments. Due to the poor performance of the GAN network in sample generation, we do not set the GAN network as the control item in the comparison experiment. Therefore, we only replace the attention module on the basis of the backbone network as shown in Figure 2, so as to eliminate the influence caused by the changes in the backbone network on the sample generation. The control items we set were: GANs, CBAM-GAN, SAGAN, and coordinate attention GAN.

In the network training, we set the training epoch as 10, the number of loops of each epoch as 10,000, and the batch size as 32. The database used is the grayscale images processed by the method shown in Section 4.1, with a total of 683 images, and all the output images are 128 in size. We replace the original resize method with the equal stretch resizing method during the training, so as to ensure the scale and shape of the input aircraft. The frequency of saving in training was 500.

It is the same as the general training steps of the GAN network; first the input noise is sent to the generator to generate the image, and then the generated image and the real image are sent to the discriminator for training. To prevent the discriminator from learning too much and causing mode collapse, we give different learning rates to the discriminator and the generator, which are 0.0005 and 0.0001, respectively. We set the mini-batch size as 5 to merge the feature maps of ten iterations, with 5 generated images and 5 real data.

### 4.3. Evaluation Factors

The original intention of using GAN to generate pseudo sample images is to make use of the limited sample resources to generate pseudo samples similarly to the original samples as far as possible, so as to expand the database. Only after the database is further expanded can it support the following tasks such as target identification and tracking. Therefore, the generated pictures must be highly similar to input samples, and the sample image must be refined enough accurately. On the other hand, in order to better train the neural network, it is necessary to generate various generated pictures. 

When evaluating the results, we used both qualitative and quantitative indicators. The qualitative index starts from the generated picture to analyze whether the generated picture meets the same characteristics and details as the sample. Quantitative indicators are used to quantify the results. Here we use two indicators, IS and FID, which are commonly used to evaluate the image quality generated by GAN.

The combination of the qualitative index and the quantitative index can well avoid the similarity of features caused in the calculation of the quantitative index while neglecting the similarity measurement of location and overall structure.

#### 4.3.1. Qualitative Evaluation

As a whole, the main features of the generated image must be as consistent as possible with the main features of the input image. For the aircraft target, the shape of the aircraft fuselage, wings, and tail as well as the position and number of engines are the characteristics that must be satisfied. As Figure 7 displays, for a civil aircraft in the database, the fuselage would be flat marked by the purple lines, the wings would be swept-back, and the front of the wings would be straight marked by the green lines. The engines are located below the wings and are usually marked by two or four red circles. 

Therefore, in addition to the symmetry of the aircraft in the generated images, attention should also be paid to the fuselage, wing edges, engine positions, and the number of engines in the qualitative evaluation of the generated images.

#### 4.3.2. Quantitative Evaluation

In the evaluation of generated pictures, there are some quantitative evaluation indicators to evaluate the diversity and quality of pictures. According to the reference [41], the index 1_NN and Kernel MMD is the most appropriate to evaluate the result of GAN models that can overcome the mode collapse, overfitting, and mode dropping situation. In order to comprehensively evaluate the results of GAN, it is necessary to prevent the influence caused by the incorrect results of one of the indicators. Therefore, we choose the inception score (IS), the Fréchet inception distance (FID), the mode score, the Kernel maximum mean discrepancy (MMD), the Wasserstein distance (WD), the and 1-Nearest Neighbor Classifier (1-NN) as the adding evaluation index of GAN models. 

Inception Score (IS)

IS is based on Google-based inception Net-v3. If the input is an image, the output of inceptionNet-v3 will be a 1000-dimension vector with each dimension of the output vector representing the corresponding probability of belonging to a certain class. IS acts as the evaluation of picture quality and diversity.

First, when we want to evaluate the quality of an image, it is important to determine which category the main object in the image belongs to. For a given generated image $x\in p_{g}$ with the main contents of the object labeled y, $p(y\left|x\right.)$ is in the case of a given x to predict the probability of type y. Therefore, we hope that the conditional probability $p(y\left|x\right.)$ can be highly predictive. Therefore, the generated images are classified using the inception network. If the inception network is able to predict the type of the image with a high probability, the generated image is of high quality; otherwise, the generated image is of low quality.

Second, the generated images should be diverse. In this case, we need to consider that the distribution of label y should have high entropy. That is, the distribution of the label y should be considered, so we need to calculate the edge probability of y.
(12)$$p(y)=\int_{z}p(y\left|x=G(z)\right.)dz$$

In the actual calculation, we use Formula (11) to replace Formula (10) to calculate the edge probability. N represents the picture amounts we input into the inception V3 network.
(13)$$\hat{p}(y)=\frac{1}{N}\sum_{i=1}^{N}p(y\left|x\right.)$$

The inception score uses KL-Divergence to evaluate the distance between the distribution probability and the edge probability of each input image. Then, the distance will be added and averaged to form the final inception score.
(14)$$IS(G)=\exp(E_{x\in p_{g}}D_{KL}(p(y\left|x\right.)\left\|p(y)\right.))$$
where $D_{KL}(•)$ is the divergence distance, which is used to measure the similarity between two probability distributions. The closer the two probability distributions are, the smaller the KL divergence distance is. $E_{x\in p_{g}}(•)$ represents the average of N images generated by GANs.

2.Fréchet Inception Distance (FID)

IS evaluates the generated images by way of classification, while FID compares the difference between generated images and real images. The goal of GANS is to make the feature distribution of the generated image close to that of the real image. The distance between two distributions can be calculated by using the Fréchet distance (FID). 

Assuming that a random variable obeys a Gaussian distribution, the distribution can be determined by means and variances. If the mean and variance of the two distributions are the same, the two distributions are the same. Therefore, the mean and the variance can be used to calculate the distance between the distributions.

For the real images with n-dimensional distribution and the feature images with generated images, the dimension of the mean is the feature dimension, and the variance is replaced by the n×n-dimensional covariance matrix.

For the real image x and generated image g, their average of feature maps are μx,μg∈ℝn×1 respectively, and covariance matrix are ∑x,∑g∈ℝn×n respectively.
(15)FID(x,y)=μx−μg22+Tr(∑x+∑g−2(∑x∑g)12)
where Tr(•) represents the sum of the elements on the diagonal of the matrix. A smaller FID value means that the distribution of x and g is closer to each other, which means that the resulting image is of higher quality and has better diversity.

3.Mode Score

Mode score is an improved version of the Inception score.
(16)$$MS(G)=\exp[E_{x\in p_{g}}(D_{KL}(p(y\left|x\right.)\left\|p(y)\right.)-D_{KL}(p(y)\left\|p(y*)\right.))]$$

p(y*) can be calculated in Formula (17), which represents the integral of the edge labeling distribution on the condition of a real sample.
(17)$$p(y*)=\int_{x}p(y\left|x=G(z)\right.)dP_{r}$$

Different from IS, the mode score can evaluate the difference between the real samples and generated pictures from the adding part $D_{KL}(p(y)\left\|p(y*)\right.)$. A higher mode score represents a better effect.

4.Kernel Maximum Mean Discrepancy (MMD)

In the calculation of Kernel MMD value, a Kernel function is first selected, which maps samples to the reproduced-kernel Hilbert Space (RKHS). RKHS has many advantages compared with Euclidean Space. The computation of the inner product of a function is complete. The smaller the MMD value, the closer the two distributions are. It can measure the advantages and disadvantages of the image generated by the model to a certain extent, which has a low calculation cost and good effect.
(18)$$MMD^{2}(P_{r},P_{g})=E_{x_{r}∼P_{r},x_{g}∼P_{g}}\left[\left\|\sum_{i=1}^{n1}k(x_{r})-\sum_{i=1}^{n2}k(x_{g})\right\|\right]$$

5.Wasserstein distance (WD)

WD calculates the distance between the two distributions, which is also applied in the evaluation of GAN models to calculate the similarity between the generated sample and real data.
(19)$$WD(P_{r},P_{g})=\min_{\omega \in R^{m\times n}}\sum_{i=1}^{n}\sum_{i=1}^{m}\omega_{ij}d(x_{i}^{r},x_{j}^{g})\;s.t.{\sum_{i=1}^{m}\omega_{i,j}=p}_{r}(x_{i}^{r})\forall i,{\sum_{i=1}^{n}\omega_{i,j}=p}_{r}(x_{j}^{g})\forall j$$

Formation 15 is the finite sample approximation of the Wasserstein distance between, used in practice. A lower Wasserstein distance represents more similarity between the two distributions.

6.1-Nearest Neighbor Classifier (1-NN)

The 1-nearest neighbor classifier is used for paired-sample testing to assess whether two distributions are identical. For two given samples, $S_{r}\sim P_{r}^{n}$ and $S_{g}\sim P_{g}^{m}$, with $\left|S_{r}\right|=\left|S_{g}\right|$, where $S_{r}$ samples from real images are positive samples and $S_{g}$ samples from generated images are negative samples, which are used to train the 1-NN classifier. When GAN obtains a good effect, and $\left|S_{r}\right|=\left|S_{g}\right|$ are very large, the 1-NN classifier should obey approximately 50% leave-one-out (LOO) accuracy. Since LOO adopts the idea of the Nash equilibrium in the calculation, it is difficult for the LOO index to be exactly 0.5. Therefore, in the experiment, the result of LOO is generally as close to 0.5 as possible.

1-NN can be used to detect overfitting and mode collapse. If the generation model produces overfitting, then the accuracy of LOO will be less than 50%. In extreme cases, if GAN remembers every sample in the real data and accurately regenerates it, the accuracy will be zero. In selecting a positive sample from the real image as the verification set, there will be a generated image with a coincident distance of 0 among the negative samples participating in the training, which will be judged as a negative category by the 1-NN classifier. Similarly, if a negative sample is selected from the generated image as the verification set, there will also be a real image that coincidences with it, thus judging it as a positive category, and the final LOO is 0. 

## 5. Results and Evaluation

This section will display, evaluate, and analyze the results of the modified shuffle attention GAN and other GAN models under the database illustrated in Section 4. 

### 5.1. Results

#### 5.1.1. Generated Picture of Different Resize Method

Shown in Figure 8a are the pictures generated by the modified shuffle attention GAN with the direct stretching resize method at the initial stage of training, and group (b) is the pictures generated by the modified shuffle attention GAN with the equal-stretch resize method proposed in this paper.

It is obvious from the image that the fake image generated by the direct stretching resize method has obvious distortion. The trunk and wings of the aircraft have different degrees of torsion. However, there is no obvious distortion in the fake image generated by the equal stretch resize method proposed in this paper. Both the trunk and wings of the aircraft are linear, the edges are smooth, and the proportion of the aircraft is well preserved.

#### 5.1.2. Generated Picture of Different GAN Models

In the experiment, we generated pictures by using GANs, SAGAN, WBAM-GAN, the coordinate attention GAN, and the modified shuffle attention GAN on the civil aircraft database mentioned above with two resize methods. For a specific network, the difference between the two resize method is similar to the results in Figure 8. Therefore, in order to draw a better comparison, we display the results of different network with the equal stretch resize method in Figure 9. Moreover, to facilitate the comparison of the proportions and shapes of airplanes in the real samples, we also selected the real sample pictures with similar angles as contrasted with the generated pictures. The planes in the real sample had different orientations, and some were individual, while others were connected to terminals.

### 5.2. Evaluation and Analyses

#### 5.2.1. Qualitative Evaluation 

First, compared with other groups, group (b) clearly has no clear shapes of the sample, while with the attention block, the generated pictures have more similar features as the samples. With the introduction of the attention module, the results of generated pictures can see a great promotion. The results depicted that with the introduction of the attention module, GANs can filter out background influences and pay more attention to key areas.

The pictures in Figure 9c are the pictures generated by SAGAN. It can be seen that some pictures in group (c) are all greatly deformed, and the fuselage and wing of the aircraft are distorted to varying degrees. The symmetry of the first and fifth slides is poor, which already have a preliminary plane shape. 

Group (d) shows the results of CBAM-GAN. Compared with SAGAN, CBAM-GAN has more stable output, and the fuselage is more detailed. However, there are still some distortions of the head of the aircraft and fuzziness of the wings’ edges. The comparison of SAGAN and CBAM-GAN reveals that the mixed-attention mechanism performed better than the single spatial attention mechanism.

Group (e) is the pseudo-sample generated by embedding the Coordinate Attention module into the backbone network. Compared with SAGAN, the fuselage and wings generated by the coordinate attention GAN are more in line with the real samples, but the description of the nose, wing edge, engine amounts, and position are not detailed enough, and the overall shape is prone to collapse.

Group (f) is the pictures generated by the modified shuffle attention GAN proposed in this paper. Compared with SAGAN and the coordinate attention GAN, the modified shuffle attention GAN is better in terms of the overall shape, proportion, and symmetry of the aircraft, as well as the detail processing of the nose, wing edge, engine position, and number. Furthermore, the interface between the terminal and the aircraft can still avoid the distortion of the aircraft itself.

#### 5.2.2. Quantitative Evaluation

Based on the pictures generated by different GAN models, we calculate the most representative index, IS and FID of the GANs, with and without equal stretch resizing to evaluate the quality of the generated pictures. Due to the bad performance of GAN displayed above, the IS and FID of GANs is not comparable with other GAN models. 

Table 1 displays the index of IS and FID of different GANs. As for IS, the modified shuffle attention GAN with the equal stretch resize method obtained the highest score of 2.114, which is 0.101 higher than SAGAN and 0.209 higher than the coordinate attention GAN. According to the meaning of IS, the results of the modified shuffle attention GAN is of the highest quality and more plentiful. Coordinate attention GAN performed the worst. Overall, the performance of networks with the equal stretch resize method is better than that of the original resize method, which is paralleled with the performance of qualitative evaluation in Figure 8.

The results of FID follow the same trend of IS. Shuffle Attention obtained the lowest FID score, which indicated that the modified shuffle attention GAN is of the best in terms of similarity to the samples. The FID score of SAGAN is 0.987 higher than the modified shuffle attention GAN and 0.237 lower than the coordinate attention GAN. 

The coordinate attention GAN’s pseudo samples were able to retain the details of the target and delineate the edges more clearly than SAGAN’s, but the quality level generated fluctuated greatly. For the coordinate attention GAN, although CA includes both channel attention and spatial attention, since CA uses the principle of two-dimensional information encoding, an error in one of the two-dimensional information encodings will result in a large error in the final generated result. Therefore, although CA can make the attention network lightweight and easily inserted into various mobile ports, it is not suitable for data expansion with high requirements for samples.

For SAGAN, using the spatial attention mechanism in each channel of the feature map after the convolution would result in an equivalent superposition of information in each channel. Once the sensing range of the task region in one of the channels is too large, it will affect the superposition of the attention region of all the channels in the later stage, thus resulting in an insufficient description of the details in the final generated picture and certain distortion of the target. Owing to its good spatial attention module, the generated pictures of SAGAN did not collapse, which is relatively smooth in the generation process. Therefore, the IS and FID of SAGAN is better than the coordinate attention GAN.

The IS and FID score of CBAM-GAN and modified shuffle attention GAN are better than SAGAN. That is, compared with the single attention module, the integrated attention module performs better, which reveals the influence of channel attention in picture generation. However, the modified shuffle attention module is better than CBAM-GAN. The modified shuffle attention GAN combines the structure of self-attention and channel attention in the backbone of Shuffle Attention Net, so that its generated picture is more detailed than SAGAN and the coordinate attention GAN. Besides, its performance of the generated pictures is as stable as SAGAN, which contributes to its good results of IS and FID.

Compared with FID and IS, which directly evaluate the quality of the generated pictures, other indicators like MMD, 1-NN can judge whether there is mode collapse and other conditions in the network results while judging the proximity between the results and the real samples.

In terms of the quality of the generated pictures, lower Kernel MMD, WD and 1-NN indexes all resemble the smaller difference between the generated pictures and real samples. From Table 2 we can clearly find that the modified shuffle attention GAN obtains the lowest index of Kernel MMD, WD, and 1-NN, which represents that the results have distributions that are closest to the real data. The results of the mode score are consistent with IS and FID in Table 1. The mode score of modified shuffle attention GAN is the highest, which also represents that the generated pictures have the closest distribution to the real samples.

In terms of mode collapse, Since SAGAN, the coordinate attention GAN, and the modified shuffle attention GAN used the same backbone network in this research, the indexes of the mode score, Kernel MMD, WD, and 1-NN showed little difference, mainly reflected in the similarity difference between the results and the real samples. CBAM-GAN adopted the original network. The 1-NN index of CBAM-GAN is higher than 0.5, which indicates there is mode collapse in CBAM-GAN. Different from the CBAM-GAN, the 1-NN index of other GAN models are close to 0.5, which represent that there are is mode collapse in the network with the adopted backbone network proposed in this paper. 

#### 5.2.3. Computational Complexity

The training time and network parameters were counted to evaluate the time complexity and space complexity of the networks. The results are shown in Table 3.

From Table 3 we can obviously acknowledge that the amount of training time of the above four GAN models are very similar. The backbone networks of SAGAN, the coordinate attention GAN, and the proposed network are the same in order to avoid the impact of the backbone network. 

In terms of time complexity, the coordinate attention GAN has the minimum training time. Compared with other GAN models, CA-Net reduces the use of convolutional layers. By encoding the feature map in two dimensions, CA-Net can efficiently reduce the calculating time. However, the two-dimension encoding also causes unstable results. The training time of the modified shuffle attention GAN is close to that of SAGAN as the addition of channel attention shows little increase in training time. In the terms of space complexity, the modified shuffle attention has fewer parameters than CBAM-GAN and the coordinate attention GAN.

The experiment results above depict that the modified shuffle attention GAN can generate high-quality and diverse images on the condition of limited size of the dataset. However, the improvement of image resolution will cause a large increase in the training time. In the future, we plan to further reduce the training time and increase the image resolution. Since different remote-sensing targets have different structural characteristics, in this paper, in order to further discuss the quality of the generated images, we conducted experiments on civil aircraft, a remote-sensing target with distinct characteristics. In the future, we will further improve the network and conduct experiments on other remote-sensing targets.

## 6. Conclusions

We introduce the modified shuffle attention module into GAN in order to generate high-quality pictures under the condition of a limited unlabeled database. In the backbone network, we added a mini-batch in the discriminator to avoid the mode collapse. In the attention module, we improved the shuffle attention network by replacing the space attention module with the modified self-attention module. Moreover, we proposed an equal stretch resize method to reduce image distortion in the process of implementation. The results of the existing image generation GAN models and the network proposed in this paper are analyzed qualitatively and quantitatively, respectively. In the qualitative analysis, pictures generated by the modified shuffle attention GAN have sharper edges and more symmetry. The number and position of engines and other details were better than those of GANs, SAGAN, CBAM-GAN, and the coordinate attention GAN. In the quantitative analysis, we selected the representative index, IS and FID, to evaluate the quality of generated pictures. The IS index of the modified shuffle attention GAN with the equal stretch resize method was the highest and reached 2.114, while the FID index was the lowest and reached 56.021, both of which indicated that the images generated by the network proposed in this paper are of higher quality and better diversity with a small amount of increased time complexity. Moreover, the mode score, kernel MMD, WD, and 1-NN also prove that the results of the GAN model we proposed have distributions that are closest to the real data. 

## Figures and Tables

**Figure 1 sensors-21-04867-f001:**
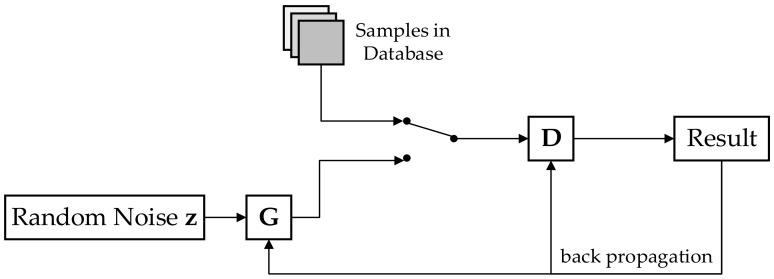
The basic structure of GAN.

**Figure 2 sensors-21-04867-f002:**
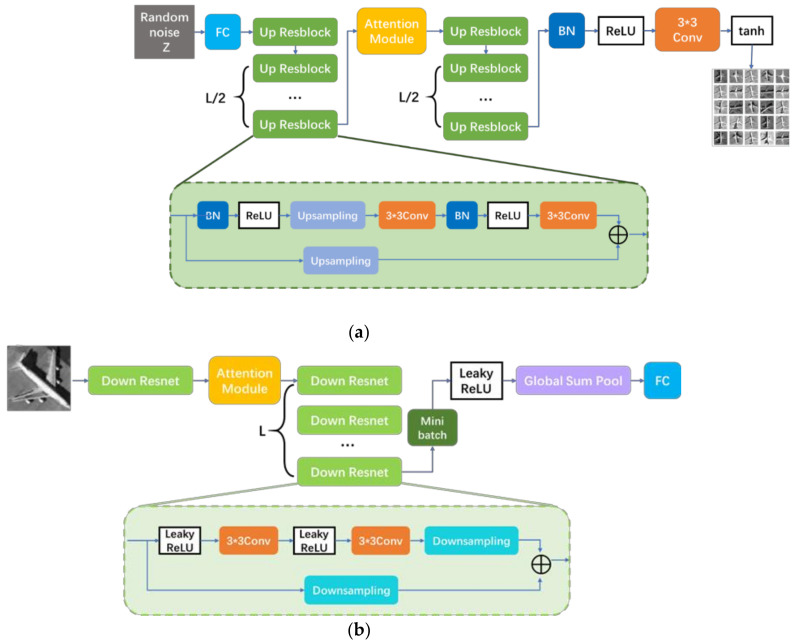
The structure of the Shuffle Attention GAN; (**a**) the structure of the generator; (**b**) the structure of the discriminator.

**Figure 3 sensors-21-04867-f003:**
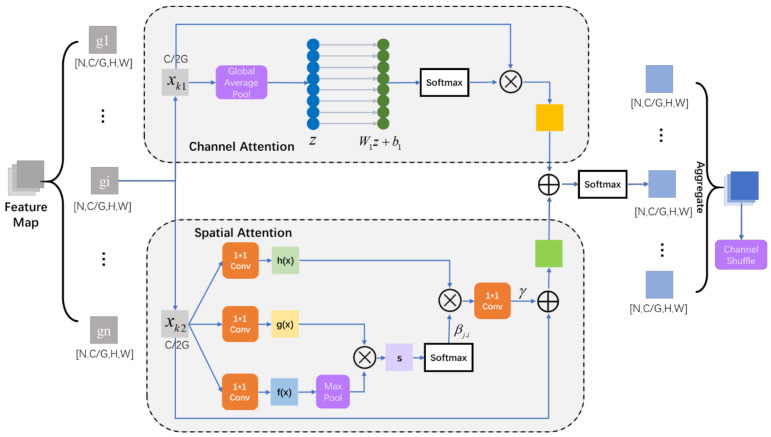
The structure of the modified shuffle attention module.

**Figure 4 sensors-21-04867-f004:**
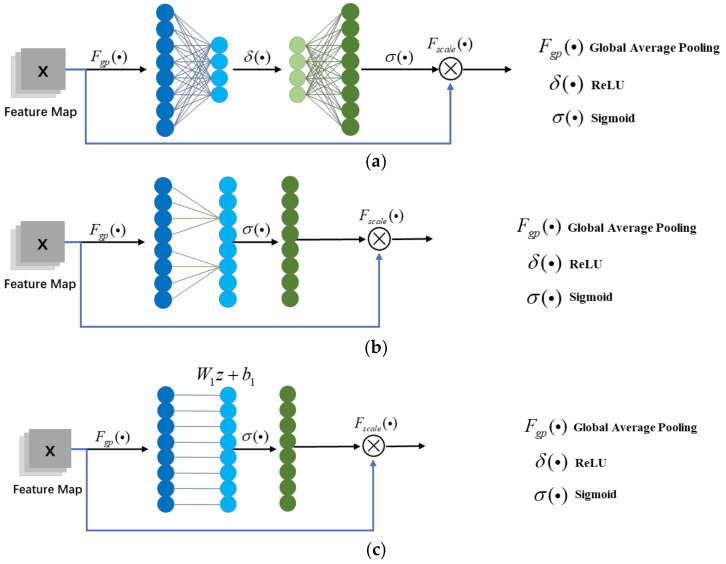
The structure of different channel attention. (**a**) The structure of SE-Net; (**b**) the structure of ECA-Net; (**c**) the structure of shuffle attention Net.

**Figure 5 sensors-21-04867-f005:**
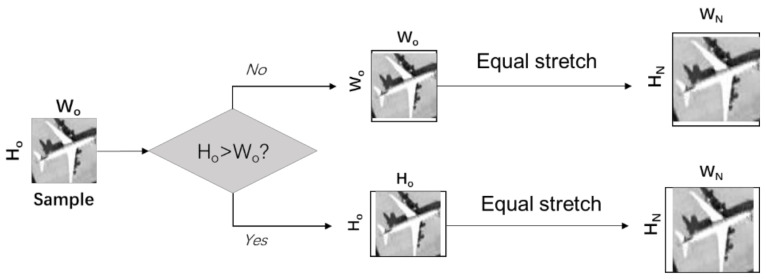
The flow diagram of the equal stretch resize.

**Figure 6 sensors-21-04867-f006:**
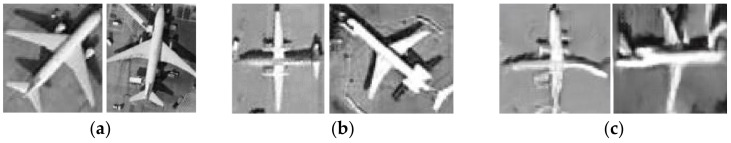
Different types of aircraft and their generated pictures. (**a**) Civil aircraft; (**b**) other kinds of aircraft; (**c**) their generated images. There is a big difference in shape between civil aircraft and other types of aircraft, which contributes to the unclassified generated images.

**Figure 7 sensors-21-04867-f007:**
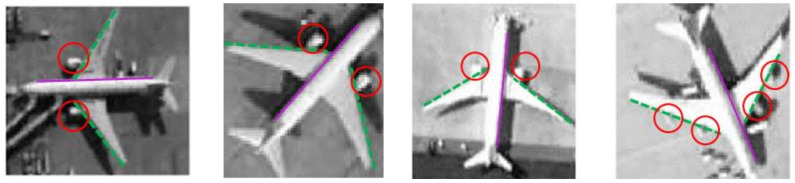
The main features of civil aircrafts.

**Figure 8 sensors-21-04867-f008:**
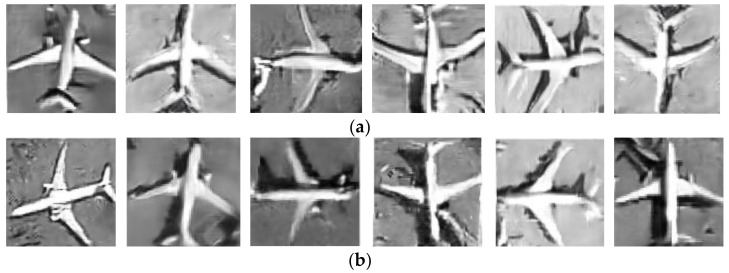
Generated pictures of modified shuffle attention GAN with and with equal stretch resize method. (**a**) Generated pictures with original resize method; (**b**) generated pictures with the equal stretch resize method.

**Figure 9 sensors-21-04867-f009:**
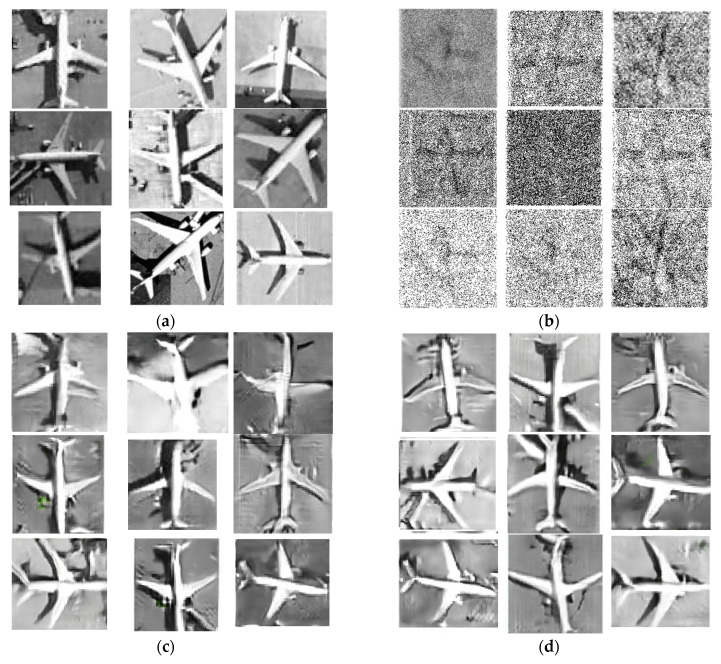
Samples of database and generated pictures of different GAN models. (**a**) Real images from the database; (**b**) generated pictures of GANs; (**c**) generated pictures of SAGAN; (**d**) generated pictures of CBAM-GAN; (**e**) generated pictures of coordinate attention GAN; (**f**) generated pictures of modified shuffle attention GAN.

**Table 1 sensors-21-04867-t001:** IS and FID scores of generated pictures of different GAN models.

Type of GAN	IS	FID
GAN + original resizing	--	--
SAGAN + original resizing	1.960	57.825
SAGAN + equal-stretch resizing	2.013	57.008
CBAM-GAN + original resizing	1.982	56.978
CBAM-GAN + equal-stretch resizing	2.021	56.615
Coordinate attention GAN + original resizing	1.864	57.582
Coordinate attention GAN + equal-stretch resizing	1.905	57.237
**Modified shuffle attention GAN(Ours)** **+ original resizing**	2.075	56.683
**Modified shuffle attention GAN(Ours)** **+ equal-stretch resizing**	**2.114**	**56.021**

**Table 2 sensors-21-04867-t002:** Other quantitative evaluation index of generated pictures of different GAN models.

Type of GAN	Mode Score	Kernel MMD	WD	1-NN
SAGAN	0.226	0.205	61.523	0.575
CBAM-GAN	0.4315	0.253	62.675	0.697
Coordinate attention GAN	0.262	0.184	58.583	0.524
**Modified shuffle attention GAN(Ours)**	**0.549**	**0.182**	**57.041**	**0.524**

**Table 3 sensors-21-04867-t003:** Training time and network parameters (Param.) of different GAN models.

Type of GAN	Training Time (h)	Param.
GAN	--	--
SAGAN	46.65	10.89M
CBAM-GAN	47.87	11.32M
Coordinate attention GAN	44.27	11.50M
**Modified shuffle attention GAN(Ours)**	**46.95**	**11.16M**

## Data Availability

1. NWPU VHR-10 dataset: Gong Cheng, Junwei Han, Peicheng Zhou, Lei Guo. Multi-class geospatial object detection and geographic image classification based on collection of part detectors. ISPRS Journal of Photogrammetry and Remote Sensing, 98: 119–132, 2014. Gong Cheng, Junwei Han. A survey on object detection in optical remote sensing images. ISPRS Journal of Photogrammetry and Remote Sensing, 117: 11–28, 2016. Gong Cheng, Peicheng Zhou, Junwei Han. Learning rotation-invariant convolutional neural networks for object detection in VHR optical remote sensing images. IEEE Transactions on Geoscience and Remote Sensing, 54(12): 7405–7415, 2016. 2. UC Merced-LandUse database: Yi Yang and Shawn Newsam, “Bag-Of-Visual-Words and Spatial Extensions for Land-Use Classification,” ACM SIGSPATIAL International Conference on Advances in Geographic Information Systems (ACM GIS), 2010.

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
