# Peer review of "Remote Sensing Image Dataset Expansion Based on Generative Adversarial Networks with Modified Shuffle Attention"

_sensors, 2021, doi:10.3390/s21144867_

Round 1

Reviewer 1 Report

The manuscript presents the authors' findings regarding applying neural networks (Generative Adversarial Networks with Modified Shuffle Attention). The overall merit is very interesting because the continuous improvement of picture recognition plays a crucial role in remote sensing. Hence, - it allows for making better (more efficient) decisions. Nevertheless, I would discuss whether the material fits the journal profile - in my opinion, it perfectly suits 'Remote Sensing' instead. The decision of qualifying the paper, however, I would leave to the 'Sensors' editorial. 

Hence, I find the paper ideally prepared. I do not have any objections neither to the form nor to the general merit. The material is convincingly presented and has suitable scientific soundness. 

I congratulate the authors on their valuable job!

Author Response

We gratefully appreciate your recognition on our manuscript and thank you very much for your good wishes!

Reviewer 2 Report

There are some comments for the authors to improve their paper which list as follows.

  1. In line 248, does the symbol N represent batch size instead of epoch, please check.
  2. In Figure 4, some lines have disappeared, please change it.
  3. In line 303, I don't find the related description in Figure 6, please check it.
  4. In line 371, please check the  product model of video card. For all I know, the RTX 3030 does not exist.
  5. In Figure 9, the (e) appeared two times, please change it.
  6. Please provide the standard deviation of quantitative evaluation.
  7. Please include a discussion about the computational complexity.

Author Response

We gratefully appreciate for your valuable suggestion, which has significantly raised the quality of the manuscript. Each suggested revision and comment brought forward by the reviewer was accurately incorporated and considered. Below the comments of are response point by point and the revisions are indicated.

1 Comment: In line 248, does the symbol N represent batch size instead of epoch, please check.

1 Reply: The mistake has been corrected in the manuscript.

2 Comment: In Figure 4, some lines have disappeared, please change it.

2 Reply: Figure 4 has been corrected in the manuscript.

3 Comment: In line 303, I don't find the related description in Figure 6, please check it.

3 Reply: Figure 6 has been changed into Figure 3 in line 303.

4 Comment: In line 371, please check the product model of video card. For all I know, the RTX 3030 does not exist.

4 Reply: The type of graphics card has been changed into RTX2070 SUPER.

5 Comment: In Figure 9, the (e) appeared two times, please change it.

5 Reply: The sequence number of the picture has been corrected.

6 Comment: Please provide the standard deviation of quantitative evaluation.

6 Reply: IS and FID(quantitative evaluation) are generated through specific evaluation networks. The standard deviation of quantitative evaluation has not been mentioned in the evaluation literature (https://arxiv.org/abs/1802.03446) and other reference of networks.

7 Comment: Please include a discussion about the computational complexity.

7 Reply: The training time and network parameters were counted to evaluate the time complexity and space complexity of the networks. Please see the revised article for more details.

Reviewer 3 Report

The article proposes generative adversarial networks with modified shuffle attention (Shuffle Attention GAN) to generate higher-quality pictures if only a limited input dataset of images is available. The results of experiments are presented which are used to demonstrate that the modified Shuffle Attention GAN proposed in this paper can generate more refined and high-quality diversified aircraft pictures with more detailed features of the object under limited datasets. The paper has some methodical, technical and presentation issues, which must be addressed before the paper could be considered for publication.

Comments:

  1. How the proposed method differs from Sequential Attention GAN (https://arxiv.org/abs/1812.08352) and Self-attention GAN (https://arxiv.org/abs/1805.08318)?
  2. The structure of the work must be improved. I suggest to add a separate Related works section, and to merge current sections 2 and 3.
  3. The overview of related works should be improved. Several recently published works presenting an important contribution to the topic should be discussed as well, including “HEMIGEN: Human embryo image generator based on generative adversarial networks” and “Combining noise-to-image and image-to-image GANs: Brain MR image augmentation for tumor detection”. The section should conclude in the summary of the advantages and limitations of GANs for dataset image augmentation, which should be used as a motivation of the current research. When discussing, outline the limitations of the previous works, which could serve as a motivation for your study.
  4. The metrics used for the evaluation of synthetic images have some deficiencies. For example, the Inception score is prone to overfitting, is agnostic to mode collapse, and is affected by image resolution. I suggest to add more performance measures to evaluate the performance of the proposed GAN: such as coverage metric and mode score, see https://arxiv.org/abs/1802.03446.
  5. How did you solve the mode collapse problem in the GAN?
  6. How did you prevent vanishing gradients while training the proposed GAN?
  7. The computational complexity of the proposed method should be evaluated.
  8. Add a Discussion section to discuss limitations of the proposed method and threats-to-validity (such as a small dataset size).
  9. Conclusions should be improved; be more specific; provide a summary of main experimental results to support your claims.
  10. Typos are present. Improve the language.

Author Response

Response to Reviewers

We gratefully appreciate for your valuable suggestion, which has significantly raised the quality of the manuscript. Each suggested revision and comment brought forward by the reviewer was accurately incorporated and considered. Below the comments of are response point by point and the revisions are indicated.

1 Comment: How the proposed method differs from Sequential Attention GAN and Self-attention GAN?

1 Reply: Sequential Attention GAN (SeqAttnGAN) is used for interactive image editing task, which can fully utilize context history to synthesize images that conform to users’ iterative feedback in a sequential fashion. Self-Attention GAN(SAGAN) introduce a self-attention module into convolutional GAN to better approximate the original image distribution by using the self-attention module to model the long-range dependencies between image regions. To be specific, Sequential Attention GAN synthesize pictures from words, while SAGAN and modified shuffle attention GAN(proposed method) generate images from existing images. Sequential Attention GAN is not discussed in the manuscript due to its different application and function. SAGAN has been discussed and compared with the proposed method in the manuscript. The biggest difference between SAGAN and the proposed network lies in the attention module. SAGAN use a single spatial attention module (self-attention module), while the proposed network uses a mixed attention module (modified shuffle attention module).

2 Comments: The overview of related works should be improved. Several recently published works presenting an important contribution to the topic should be discussed as well, including “HEMIGEN: Human embryo image generator based on generative adversarial networks” and “Combining noise-to-image and image-to-image GANs: Brain MR image augmentation for tumor detection”. The section should conclude in the summary of the advantages and limitations of GANs for dataset image augmentation, which should be used as a motivation of the current research. When discussing, outline the limitations of the previous works, which could serve as a motivation for your study.

2 Reply: The overview of related works has been improved. Please see the revised article for more details. We have studied carefully the reference you mentioned. However, the research direction of the reference mentioned above is not consistent with that of the manuscript.

3 Comments: The metrics used for the evaluation of synthetic images have some deficiencies. For example, the Inception score is prone to overfitting, is agnostic to mode collapse, and is affected by image resolution. I suggest to add more performance measures to evaluate the performance of the proposed GAN: such as coverage metric and mode score, see https://arxiv.org/abs/1802.03446.

3 Reply: We also use Fréchet Inception Distance (FID) as the quantitative evaluation index to evaluate the generated images of different GAN models. FID compare the difference between generated images and real images, which can make up for the weakness of IS.

4 Comments: How did you solve the mode collapse problem in the GAN?

4 Reply: Generally, mode collapse is closely related with discriminator. Discriminator an only process one sample independently at a time which contributes to the lack of information coordination between samples. Mini-batch discriminator is an effective solution to avoid mode collapse (1606.03498.pdf (arxiv.org)). The simplified version of mini-batch (1710.10196.pdf (arxiv.org)) has been added after the last down-Reset layer to avoid mode collapse. Please see the revised article for more details.

5 Comments: How did you prevent vanishing gradients while training the proposed GAN?

5 Reply: We reduce the use of Softmax activation functions in the backbone network and use ReLU, leaky ReLU instead. Moreover, we also introduce BN (batch normal) to avoid the vanishing gradients.

6 Comments: The computational complexity of the proposed method should be evaluated.

6 Reply: The training time and network parameters were counted to evaluate the time complexity and space complexity of the networks. Please see the revised article for more details.

7 Comments: Add a Discussion section to discuss limitations of the proposed method and threats-to-validity (such as a small dataset size).

7 Reply: The discussion section has been added in section 5. Please see the revised article for more details.

8 Comments: Conclusions should be improved; be more specific; provide a summary of main experimental results to support your claims.

8 Reply: The conclusion has been improved. Please see the revised article for more details.

Reviewer 4 Report

In this paper, the authors proposed a modified GAN by using the shuffle attention net as backbone of GAN.  I can note that this is a good research, because the experimental results demonstrated the high performance of the proposed method. 

I have one question:

In fact, there are many types of remote sensing data. However, the authors used only image based datasets. Since the title is "remote sensing dataset"; therefore, I suggest that the authors should also apply the proposed method to some non-image based remote sensing datasets to demonstrate the effectiveness. 

Author Response

We gratefully appreciate for your valuable suggestion, which has significantly raised the quality of the manuscript. The manuscript is aim at enlarge the database of remote sensing images. Therefore, we have changed the title “Remote Sensing Dataset Expansion Based on Generative Adversarial Networks with Modified Shuffle Attention” into “Remote Sensing Image Dataset Expansion Based on Generative Adversarial Networks with Modified Shuffle Attention”. And we will carry on relevant research on the application of the method into other types of remote sensing database.

Round 2

Reviewer 3 Report

Unfortunately, the authors did only minor corrections and my main concerns remain unaddressed. Specifically, the authors have failed to outline the novelty of this manuscript. The proposed GAN architecture is very similar to the architectures proposed in several other papers, which also included “self-attention” and “mixed-attention” modules. The metrics used for the evaluation of synthetic images are deficient and prone to overfitting, agnostic to mode collapse problem, and are affected by image resolution. The authors have ignored the suggestion to add more informative performance measures to evaluate the performance of the proposed GAN such as coverage metric and mode score. The dataset used for experiments is limited, and the description of the experimental settings lacks of necessary detail for reproducibility. As a result, I am recommending the rejection of this paper.

Author Response

Dear reviewer:

We would like to express our sincere apologies for our oversight in the first round. For the questions and valuable suggestions you raised in the first and second rounds, we will elaborate in detail below.

  • As for the novelty of the manuscript, we adopt the idea of shuffle attention network but we modified the structure of origin shuffle attention network. The main out research and the contributions of this paper are as follows: (from line 144-169 in introduction)
  1. We adjust the structure of shuffle attention network by replacing the original spa-tial attention module with modified self-attention network to obtain better spatial attention. Then we proposed modified shuffle attention GAN by introducing the modified shuffle attention net into GAN.
  2. We introduce the mini-batch into the backbone network to avoid mode collapse in GAN under small dataset.(line 315-328) .In the later experiment, the 1-NN index verified the effect brought by this introduction.
  3. We introduce coordinate attention network into GAN to form coordinate attention GAN as contrast with shuffle attention GAN. We integrate the civil aircraft in dataset NWPUVHR-10 and UC merced Landuse and select the input pictures to ensure the purity of input type of aircraft.
  4. We proposed equal-stretch resize method to avoid the distortion of image. We al-so do contrast experiments on the modified shuffle attention GAN with and with-out equal-stretch resize method. The results shows that the equal-stretch resize method can apparently avoid the distortion.
  5. We applied qualitative and quantitative evaluation index to judge the quality of generated pictures. The qualitative evaluation consists of the appearance of fuselage, the amount and location of engine, the symmetry of aircraft. The representative quantitative evaluation index consists of inception score (IS) and fréchet inception distance (FID), which are the most representative index for evaluation of the generated pictures. mode score, Kernel Maximum Mean Discrepancy (MMD), wasserstein distance (WD and 1- Nearest Neighbor Classifier (1-NN). We make comparison and do evaluation on the results of modified shuffle attention GAN and existing image generation GAN. The experiments shows that the modified shuffle attention GAN performed better than GAN, SAGAN, CBAM-GAN, coordinate attention GAN.

Moreover, the specific structural changes of self-attention module is: we adjust the structure of self-attention module in our proposed network by placing a max-pooling in one branch of convolution to reduce the computational complexity caused by cross correlation calculation. (see Figure 3 spatial attention module).

  • We added more evaluation index as the quantitative evaluation indexes in section 4 from line 558~599. The details of adding index are as follows:

According to the reference (https://arxiv.org/abs/1802.03446) and reference (https://arxiv.org/pdf/1806.07755.pdf), the index 1_NN and Kernel MMD is the most appropriate to evaluate the result of GAN models for which can overcome the mode collapse ,overfitting and mode dropping situation. In order to comprehensively evaluate the results of GAN, it is necessary to prevent the influence caused by the incorrect results of one of the indicators. Therefore, we choose Mode score, Kernel Maximum Mean Discrepancy (MMD), Wasserstein distance (WD) and 1-Nearest Neighbor Classifier (1-NN) as the adding evaluation index of GAN models.

  1. Mode score is a modified version of Inception score, which adding the evaluation of the difference between the real sample and generated pictures.
  2. Kernel MMD select a Kernel function in the calculation, which maps samples to the reproduced-kernel Hilbert Space (RKHS). The smaller the MMD value, the closer the two distributions are. It can measure the advantages and disadvantages of the image generated by the model to a certain extent, which has low calculation cost and good effect.
  3. WD calculates the distance between the two distributions, which is also applied in the evaluation of GAN models to calculate the similarity between the generated sample and real data.
  4. 1-NN can be used to detect overfitting and mode collapse. If the generation model produces overfitting, then the accuracy of LOO will be less than 50%. In extreme cases, if GAN remembers every sample in real data and accurately regenerates it, the accuracy will be zero. Selecting a positive sample from the real image as the verification set, there will be a generated image with a coincident distance of 0 among the negative samples participating in the training, which will be judged as a negative category by the 1-NN classifier. Similarly, if a negative sample is selected from the generated image as the verification set, there will also be a real image that coincidences with it, thus judging it as a positive category, and the final LOO is 0. Since LOO adopts the idea of Nash equilibrium in the calculation, it is difficult for the LOO index to be exactly 0.5. Therefore, in the experiment, the result of LOO is generally close to 0.5 as far as possible

The result of the evaluation indexes is shown in Table 2. (see line 709-732). Here we display the main conclusion of the result.

Among the GAN models, our GAN model gets the highest mode score, lowest Kernel MMD and WD score, which depicted that the generated pictures represent the generated pictures have the closest distribution to the real samples. Moreover, the 1-NN score of our GAN model is the closet to 0.5, which represent there is no overfitting and mode collapse in our model.

  • We do our experiments on the database of civil aircraft in NWPUVHR-10 and UC Merced Land-use. In this manuscript, we are aiming at expanding remote sensing dataset. In this case, the input samples will be limited. Since different remote sensing targets have different structural characteristics, in this paper, in order to further discuss the quality of the generated images, we conduct experiments on civil aircraft, a remote sensing target with distinct characteristics. In the future, we will further improve the network and conduct experiments on other remote sensing targets.

The specific steps of the experiment are as follows:

  1. Pretreatment: we select the civil images and transferred the samples into gray images. After that, we use the equal-stretch resizing method proposed in this manuscript to uniform the input size of samples to avoid image distortion.(line 438-462 section 4.1)
  2. In the network training, we set the training epoch as 10, the number of loops of each epoch as 10,000, and the batch size as 32. The database used is the grayscale images processed by the method shown in Section 4.1, with a total of 683 images, all the output images are 128 in size. We replace the original resize method with equal-stretch resizing method during the training, so as to ensure the scale and shape of input aircraft. The frequency of saving in training was 500. (line 475-480 section 4.2)
  3. It is the same as the general training steps of GAN network, first the input noise is sent to the generator to generate the image, and then the generated image and the real image are sent to the discriminator for training. To prevent the discriminator from learning too much and causing the mode collapse, we give different learning rates to the discriminator and the generator, which are 0.0005 and 0.0001 relatively. We set the mini-batch size as 5 to merge the feature maps of ten iteration, with 5 generated images and 5 real data. (see line 416-422)
  4. The results of the discriminator are transmitted back and the loss function is used to update the discriminator and generator, respectively, until the loop of 10 epochs is completed.
  5.  

Reviewer 4 Report

The authors have responded my questions, I have no other extended comments.

Author Response

Thank you for your valuable recognition of this article! The article has been further modified, please see the attachment.